# Binge-Watching and Mental Health Problems: A Systematic Review and Meta-Analysis

**DOI:** 10.3390/ijerph19159707

**Published:** 2022-08-06

**Authors:** Zainab Alimoradi, Elahe Jafari, Marc N. Potenza, Chung-Ying Lin, Chien-Yi Wu, Amir H. Pakpour

**Affiliations:** 1Social Determinants of Health Research Center, Research Institute for Prevention of Non-Communicable Diseases, Qazvin University of Medical Sciences, Qazvin 34197-59811, Iran; 2Departments of Psychiatry and Neuroscience, The Child Study Center, School of Medicine, Yale University, New Haven, CT 06511, USA; 3Institute of Allied Health Sciences, College of Medicine, National Cheng Kung University, University Rd., Tainan 701401, Taiwan; 4Department of Pediatrics, E-Da Hospital, Kaohsiung 82445, Taiwan; 5School of Medicine, College of Medicine, I-Shou University, Kaohsiung 82445, Taiwan; 6Department of Nursing, School of Health and Welfare, Jönköping University, 553 18 Jönköping, Sweden

**Keywords:** binge-watching, depression, loneliness, sleep problems, anxiety, stress, addictive behaviors, Internet use addiction

## Abstract

Background: Binge-watching, the viewing of online videos or streamed content, may be associated with different types of mental health problems. The present study aimed to investigate the associations between binge-watching and five mental health concerns including depression, loneliness, sleep problems, anxiety, and stress. Methods: Academic databases of PubMed, Scopus, Web of Science, ProQuest, PsycINFO, and Psych Articles were systematically searched through February of 2022. The Newcastle–Ottawa Scale was used to assess the methodological quality. A meta-analysis was performed on Fisher’s z values as effect sizes, using a random effect model. Publication bias, small study effect, and moderators in this association were assessed. Results: Binge-watching was significantly associated with the five types of mental health concerns with the most robust correlations found with stress (0.32) and anxiety (0.25). Stronger associations between binge-watching and two types of mental health problems (depression and sleep problems) were found during the COVID-19 pandemic than before the pandemic. Moreover, stronger associations between binge-watching and two types of mental health problems (stress and sleep problems) were found in developing countries than in developed countries. Conclusions: The associations between binge-watching and mental health concerns were significant and positive. Programs and interventions to reduce binge-watching should be considered and tested.

## 1. Introduction

Binge-watching is a relatively new phenomenon involving watching online videos or streamed content (e.g., on Netflix) [1]. More specifically, the watching of traditional television (TV) programming has been historically restricted by the timetable or schedule of each program (e.g., each episode of a TV series is released weekly and thus individuals need to follow the schedule to watch a series). However, such restrictions in TV-watching have largely been removed nowadays because online videos or streaming channels can provide an entire season (or several seasons) of a TV series on demand [2]. With this flexibility, some people may spend substantial time watching episodes in rapid succession, and such marathon-like watching has been termed as binge-watching [3,4,5].

Although binge-watching has become a global norm and some people use this as one type of their leisure activities [1,2,5], some evidence suggests that this pattern of viewing may be linked to health problems including both physical and mental health concerns [5,6,7]. For example, binge-watching may lead to sedentary behaviors because of the long times spent sitting and contribute to unhealthy lifestyles [7]. Repetitive binge-watching may result in sleep problems because of disruptions to circadian rhythms [6]. Binge-watching may also jeopardize the individuals’ social relationships as people may neglect important social activities and reduce their social interactions [5]. Therefore, there is a growing interest among health care providers and health care researchers to understand the impacts of binge-watching.

Binge-watching may have increased during the COVID-19 pandemic [8]. With the use of some infection–prevention policies (e.g., quarantines and lockdown) [9,10,11], problematic use of the Internet has been a particular concern during the COVID-19 pandemic [12,13], as has binge-watching [14,15]. The literature suggests that isolation during the COVID-19 pandemic may be a potential reason for people engaging in binge-watching. During the pandemic, people had limited daily activities for coping with their psychological needs, and binge-watching was one option [14,15]. In other words, psychological distress (e.g., fear and anxiety) experienced during the COVID-19 pandemic may have influenced binge-watching behaviors [16,17,18]. As a result, a vicious cycle could have developed as people may have used binge-watching to meet their psychological needs to cope with the COVID-19 pandemic. Consequently, people may have experienced adverse effects relating to binge-watching. Moreover, streaming channels often had promotions during the COVID-19 pandemic [8,19], and this practice may have promoted binge-watching and its correlates.

Associations between binge-watching and mental health have been empirically studied. Such associations appear to be supported by the Uses and Gratification Theory. Specifically, people may binge-watch to satisfy their mental health needs [1]. Additionally, the Uses and Gratification Theory indicates that people may receive gratification, particularly if the activity is planned [4]. However, binge-watching may be an unplanned behavior (i.e., individuals attempt not to watch too much but have difficulty in ceasing the behavior). Therefore, such unplanned binge-watching may result in regret and the worsening of mental health concerns [4]. As such, binge-watching may constitute a specific form of problematic use of the Internet, consistent with the interaction of person, affect, cognitive, and execution (I-PACE) theory. However, to the best of the present authors’ knowledge, no prior systematic review involving a meta-analysis has been conducted to synthesize the data regarding the associations between binge-watching and mental health. One recent systematic review on binge-watching has summarized the evidence regarding (i) reasons for binge-watching and (ii) negative measures associated with binge-watching [1]. In this systematic review, motivations (e.g., relating to psychological needs, social nature, fear of missing out, escaping from reality, and social engagement) and personality features (e.g., neurotic, less agreeable, less openness to new experiences) were linked to binge-watching [1]. Moreover, potential health risks resulting from binge-watching include developing behavioral addictions, sleep problems, sedentary behaviors, and psychological distress (e.g., depression, anxiety, loneliness, and stress). Although the systematic review summarized the issues relating to binge-watching [1], it lacked quantitative evidence to support associations between binge-watching and mental health problems. Moreover, this prior systematic review did not consider the impacts of the COVID-19 pandemic on binge-watching. In order to consider evidence in a quantitative manner, there is a need to use a systematic review with a meta-analysis to further understand the associations between binge-watching and mental health problems, with the consideration of the possible changes occurring during the COVID-19 pandemic.

### 1.1. Study Aims

In order to address a literature gap, the present systematic review and meta-analysis was conducted using rigorous study methodology with the main aim of examining the data linking binge-watching to mental health concerns including depression, loneliness, sleep problems, anxiety, and stress. Secondary study purposes were to investigate whether the associations between binge-watching and mental health were different before and during the COVID-19 pandemic and between developing and developed countries given the potential differences in binge-watching and mental health during these time periods and in these jurisdictions. The merits of the present meta-analysis include using published references to understand the associations between binge-watching and mental health problems. This is particularly important given the changes in technology that have facilitated binge-watching behaviors and mental health concerns that may relate to screen media activity and specific forms such as binge-watching. Moreover, meta-analyses quantify and accumulate existing evidence and can provide clear and robust information regarding associations.

#### 1.1.1. Primary Aim

Investigating the association of binge-watching with five types of mental health concerns including depression, loneliness, sleep problems, anxiety, and stress.

#### 1.1.2. Secondary Aims

i.Assessment of heterogeneity and its possible sources in associations between binge-watching and mental health problems;ii.Moderator analyses to determine influential variables in associations between binge-watching and mental health problems;iii.Subgroup analyses to investigate, for example, the potential effects of the COVID-19 pandemic on associations between binge-watching and mental health problems. Specifically, we defined a study as having been conducted during the COVID-19 pandemic when its data collection occurred after December 2019.

## 2. Materials and Methods

### 2.1. Protocol and Registration

Preferred Reporting Items for Systematic Reviews and Meta-Analyses guideline version 2020 [20] was used to report this systematic review and meta-analysis. The protocol of this study was registered in PROSPERO (international prospective register of systematic reviews) under the PROSPERO-ID of CRD42022309716 [21].

### 2.2. Eligibility Criteria

Observational studies including longitudinal (i.e., having two or more assessment time points in the study), case-control (i.e., having one group with binge-watching concerns and another group without in the study), and cross-sectional studies (i.e., having only one assessment time point in the study) published in English through February 2022 were included if relevant data regarding binge-watching and associations with selected mental health problems (depression, loneliness, sleep problems, anxiety, and stress) were reported. No limitations were exerted regarding the participants’ characteristics. For the meta-analysis, studies were included that reported a correlation score between binge-watching and mental health problems including depression, anxiety, insomnia, loneliness, and stress. To be included in the meta-analysis, the selected mental health problems needed to have been assessed using a valid and reliable measure.

### 2.3. Information Sources

The review team searched academic databases, namely, PubMed, Scopus, Web of Science, ProQuest, PsycINFO, and Psych Articles from inception through February 2022. To conduct a more comprehensive search, gray literature including Google Scholar, ProQuest thesis and dissertation, conference papers, and the reference lists of included papers were searched.

### 2.4. Search

Search terms (“Binge-watching” OR “Binge-viewing” OR “Marathon viewing” OR “Marathon watching” OR “Media marathoning” OR “Increased viewing” OR “Excessive viewing” OR “Problematic viewing” OR “TV series” OR “TV shows” OR “TV dramas”) AND (“depress*” OR “anxiety” OR “social phobia” OR “agoraphobia” OR “panic disorder” OR “obsessive-compulsive disorder” OR “sleeplessness” OR “chronic insomnia” OR “insomniac” OR “insomnia” OR insomni* OR “sleep initiation” OR “maintenance disorders” OR “mental health” OR “Mental Hygiene” OR “Psychological well-being”). Search syntax was adopted for each database based on their advanced search attributes. The flow of the study selection process is summarized in the PRISMA (2020) flowchart [20]. E.J. and Z.A. contributed in developing the search syntax and conducting the search of academic databases.

### 2.5. Study Selection

The titles and abstracts of all of the retrieved papers were screened based on the inclusion criteria. Next, full texts of potentially relevant studies based on the criteria above-mentioned were assessed, and relevant studies were selected.

### 2.6. Data Collection Process

An Excel datasheet was designed to extract the data. Two reviewers (E.J. and Z.A.) extracted the data independently using this predefined sheet.

### 2.7. Data Items

Data items included the first author, year of publication, study design, country, country developmental status (based on latest world bank data), sample size, target population, number of men and women, the names of measurements for mental health and binge-watching, age range or mean age, and effect size estimate of binge-watching and mental health. We included the variable of country developmental status in the meta-analysis because countries at different levels of development may exhibit different patterns of binge-watching behaviors and mental health problems (e.g., mental health problems may be more serious in developing countries than in developed countries because developing countries may not have as well-established mental health care systems).

### 2.8. Risk of Bias in Individual Studies

The Newcastle–Ottawa Scale (NOS) was used to assess the risk of bias within the included studies. This checklist evaluates the methodological quality of observational studies in the following three sections: selection, comparability, and outcome [22]. The maximum acquirable score on the NOS checklist is 9 for each study. Studies with less than five points were classified as having a high risk of bias [22]. Methodological quality status was not considered as an eligibility criterion. However, the effect of methodological quality on the pooled effect size was assessed in the subgroup analysis and meta-regression.

### 2.9. Summary Measures

Pearson’s correlation coefficients were used to estimate the effect sizes in the analyses of associations between binge-watching and mental health problems. Because of the potential instability of variance, Pearson’s r correlation coefficients were transformed to Fisher’s z-statistics [23,24]. Consequently, all analyses were performed using Fisher’s z values as effect sizes (ESs). The transformation of the Pearson r correlation coefficients to Fisher’s z-values was conducted using the formula of z = 0.5 × ln[(1 + r) − (1 − r)]. The standard error of z was calculated based on the following formula: SEz = 1/√(n − 3) [25]. Therefore, the selected key measure for the current meta-analysis was the Fisher’s z-score and its 95% CI. The magnitude of a Fisher’s z value is defined as follows: at 0.1 is weak, between 0.11 and 0.29 is weak to moderate, 0.3 is moderate, 0.31–0.49 is moderate to strong, and ≥0.5 is strong [26].

### 2.10. Synthesis of Results

The data analysis was conducted using STATA software version 14.0. (College Station, TX, USA). As the included studies were taken from different populations, random effect models were used to conduct the meta-analysis. Random effect models can estimate within- and between-study variances [27]. The I^2^ index estimated the heterogeneity across the included studies. Heterogeneity was interpreted as mild with an I^2^ < 25%, moderate with 25 < I^2^ < 50%, severe with 50 < I^2^ < 75%, and highly severe with I^2^ > 75% [28].

### 2.11. Risk of Bias across Studies

The probability of publication bias was assessed using funnel plots and Egger’s test [29]. The Jackknife method was used to assess the small study effects [30].

### 2.12. Additional Analyses

To assess the moderator variables and find the main source of heterogeneity, subgroup analysis or meta-regression was conducted. Subgroup analysis is arguably the most widely used method to examine sources of heterogeneity in meta-analyses and involves the division of studies into two or more subgroups based on the selected variables. Significant differences in the pooled effect sizes among these subgroups are then examined [31,32]. Meta-regression is arguably a more sophisticated method than subgroup analysis for exploring heterogeneity and has the potential advantage of efficiently allowing the evaluation of one or more covariates simultaneously [33]. Two indicators were assessed in the meta-regression including the tau-square value and I^2^ residual. The Tau-squared (τ^2^ or Tau^2^) is an estimate of the between-study variance in a random-effects meta-analysis [34]. The I^2^ residual is an indicator showing the effect of a selected variable on the observed heterogeneity. Higher values of residual variance indicate that the examined variable is not an important source of heterogeneity [35]. Notably, the threshold for deciding on the significance of a *p*-value in meta-regression depends on the number of studies: when there are fewer than 10 studies, the threshold is 0.20; between 10 and 20 studies, the threshold is 0.15; and above 29 studies, the threshold is 0.10 [25,33].

## 3. Results

### 3.1. Study Screening and Selection Process

The initial search retrieved 301 studies, and 44 duplicated manuscripts were removed. A further 257 manuscripts were screened based on the title and abstract. No references were added after conducting a search in Google Scholar, ProQuest thesis and dissertation, conference papers, and the reference lists of included papers. Finally, the full texts of 141 manuscripts were reviewed for eligibility criteria. In this process, 13 peer-reviewed manuscripts and three theses were included in the meta-analysis. Figure 1 shows the search process based on the PRISMA flowchart.

### 3.2. Study Description

The 16 included studies (four from the USA, three from the Netherlands, and one from each of the following regions: Pakistan, Poland, Taiwan, Germany, Belgium, United Arab Emirates, Italy, Turkey, and India) involved 8077 total participants. The age range of the participants was 18–68 years and the participants were predominantly (62.97%) females. The sample size varied between 38 and 1488 participants. All studies were cross-sectional. The mental-health-related factors assessed included depression (n = 9), loneliness (n = 6), insomnia (n = 5), anxiety (n = 6), and stress (n = 3). Five studies collected data during the COVID-19 pandemic. A summary of the characteristics of the included studies are provided in Table 1.

### 3.3. Methodological Quality Appraisal

All studies met the criteria for the validated measurement tools, used clear statistical tests to analyze, and self-reported the assessment of outcomes. The most common problems were not reporting the justification or sample size calculation and the number of non-respondents. Overall, 62.5% of the included studies were ranked as having low methodological quality (NOS scores < 5). The total quality scores are provided in Table 1 and Figure 2.

### 3.4. Outcome Measures

Binge-watching was positively and significantly associated with different mental health problems including stress (ES: 0.32), loneliness (ES: 0.19), anxiety (ES: 0.25), depression (ES: 0.14), and insomnia (ES: 0.16) (Table 2). The results of the pooled estimates for associations between binge-watching and different mental health problems including moderator analysis, publication bias assessment, and sensitivity analysis are reported.

#### 3.4.1. Associations between Binge-Watching and Depression

Nine studies reported the associations between binge-watching and depression. The pooled estimated ES showed weak correlations between binge-watching and depression with a Fisher’s z-score of 0.14 [95% CI: 0.04 to 0.24, I^2^= 90.6%; Tau^2^ = 0.02]. Figure 3 provides the forest plot regarding the association between binge-watching and depression. Publication bias seems probable in associations between binge-watching and depression based on the Egger’s test (*p* = 0.09) and asymmetrical funnel plot (Figure 4). In this regard, the fill-and-trim method was used to correct the probable publication bas, but no study was imputed and the publication bias was excluded. The sensitivity analysis showed that the pooled ES was not affected by the effect of a single study (Figure 5).

The subgroup analysis (Table 3) showed that the association between binge-watching and depression was significantly different before and during the COVID-19 pandemic (0.05 vs. 0.28); not significantly different between the low and high risk of bias (0.12 vs. 0.15); and significantly different between developing vs. developed countries (0.20 vs. 0.12). Based on the univariate meta-regression analysis (Table 4), during the pandemic, the participants’ mean age (in years) and female gender were statistical predictors with respect to the association between binge-watching and depression. Further multivariate meta-regression analysis showed that these variables explained 85.92% of variance in the association. During the COVID-19 pandemic, the association between binge-watching and depression was 0.25 higher compared to before the COVID-19 pandemic. Increases in the participants’ mean age was associated with small decreases in this association (−0.04 decrease of Fisher’s z-score by a one year increase in the participants’ mean age). Increases in the percentage of female participants were associated with small increases in this association (0.008 increase of Fisher’s z-score by a 1 percent increase in the female participants).

#### 3.4.2. Associations between Binge-Watching and Anxiety

Six studies reported on the associations between binge-watching and anxiety. The pooled estimated ES showed a weak to moderate association between binge-watching and anxiety with a Fisher’s z-score of 0.25 [95% CI: 0.14 to 0.35, I^2^ = 89.7%; Tau^2^ = 0.01]. Figure 6 provides the forest plot regarding the association between binge-watching and anxiety. Publication bias was not probable based on the Egger’s test (*p* = 0.76) and symmetrical funnel plot (Figure 7). Sensitivity analysis showed that the pooled ES was not affected by the effect of a single study (Figure 8).

Subgroup analyses (Table 3) showed that associations between binge-watching and anxiety were not significantly different before and during the COVID-19 pandemic (0.21 vs. 0.29); not significantly different between low and high risk of bias (0.20 vs. 0.27); and significantly different between developing vs. developed countries (0.34 vs. 0.22). Based on univariate meta-regression analyses (Table 4), the participant’s mean age and proportions of female participants were statistical predictors in associations between binge-watching and anxiety. A multivariate meta-regression analysis showed that the participants’ mean age was a significant moderator in the association between binge-watching and anxiety, and age explained 99% of variance in this association. Each year of increasing age was associated with 0.19 of the Fisher’s z-score increase in this association.

#### 3.4.3. Associations between Binge-Watching and Stress

The studies conducted during the COVID-19 pandemic reported associations between binge-watching and stress. The pooled estimated ES showed a moderate association between binge-watching and stress with Fisher’s z-score of 0.32 [95% CI: 0.20 to 0.44, I^2^ = 86.9%; Tau^2^ = 0.01]. Figure 9 provides the forest plot regarding the association between binge-watching and stress. The probability of publication bias was ruled out in the association between binge-watching and stress based on the Egger’s test (*p* = 0.63) and symmetrical funnel plot (Figure 10). The sensitivity analysis showed that the pooled ES was not affected by the effect of a single study (Figure 11).

Due to low number of total studies, the subgroup analysis was not conducted for associations between binge-watching and stress. Based on univariate meta-regression analyses (Table 4), the country (Pakistan: 0.41; USA: 0.33; and Italy: 0.22), NOS score, and proportion of female participants were statistical predictors in associations between binge-watching and stress. Further multivariate meta-regression analyses were not possible due to an insufficient number of studies.

#### 3.4.4. Associations between Binge-Watching and Loneliness

Six studies reported on the associations between binge-watching and loneliness. The pooled estimated ES showed a weak association between binge-watching and loneliness with a Fisher’s z-score of 0.19 [95% CI: 0.13 to 0.25, I^2^ = 65.2%; Tau^2^ = 0.01]. Figure 12 provides the forest plot regarding the association between binge-watching and loneliness. Publication bias seems probable based on the Egger’s test (*p* = 0.08) and symmetrical funnel plot (Figure 13). However, using the fill-and-trim method to correct probable publication bias, no study was imputed, and publication bias was excluded. Sensitivity analysis showed that the pooled ES was not affected by the effect of a single study (Figure 14).

Based on the subgroup analysis (Table 3) and meta-regression (Table 4), none of the examined variables influenced the pooled ES or heterogeneity in the association between binge-watching and loneliness.

#### 3.4.5. Associations between Binge-Watching and Insomnia

Five studies reported on the associations between binge-watching and insomnia. The pooled estimated ES showed a weak association between binge-watching and insomnia with a Fisher’s z-score of 0.16 [95% CI: 0.03 to 0.30, I^2^ = 88.4%; Tau^2^ = 0.02]. Figure 15 provides the forest plot regarding the association between binge-watching and insomnia. Publication bias was not probable based on the Egger’s test (*p* = 0.34) and symmetrical funnel plot (Figure 16). Sensitivity analyses showed that the pooled ES was not affected by the effect of a single study (Figure 17).

Subgroup analysis (Table 3) showed that the association between binge-watching and insomnia was significantly stronger during versus before the COVID-19 pandemic (0.32 vs. 0.11). Additionally, this association was numerically higher in low vs. high risk of bias studies and in developing vs. developed countries, but the differences were not statistically significant. Based on the univariate meta-regression analyses (Table 4), all of the examined variables including country, methodological quality score, timeframe with respect to the COVID-19 pandemic, the participants’ mean age, and the proportion of female participants were statistical predictors in the association between binge-watching and insomnia. Further multivariate meta-regression analysis was not possible due to an insufficient number of studies.

## 4. Discussion

To the best of the present authors’ knowledge, this is the first study that summarized and synthesized, using meta-analytic methodologies, the evidence regarding the associations between binge-watching and mental health problems. The evidence was derived from 8077 adults (62.97% females) across three continents (including Asia, Europe, and the Americas). Moreover, the associations were further explored to determine whether they differed before and during the COVID-19 pandemic and in developing versus developed countries. Through a thorough literature search and a robust methodological approach, the meta-analytic results showed that binge-watching was positively and significantly associated with multiple mental health problems including stress (ES: 0.32), loneliness (ES: 0.19), anxiety (ES: 0.25), depression (ES: 0.14), and insomnia (ES: 0.16).

The present findings indicate that the numerically most robust correlations were between binge-watching and stress (0.32) and anxiety (0.25), respectively. These correlations resonate with the Uses and Gratification Theory [55,56]. That is, individuals may use media to satisfy their needs including psychological ones. Therefore, individuals may binge-watch to cope with troublesome and disturbing life experiences that may generate stressful and anxious feelings, or individuals may encourage their friends to watch the same series and such binge-watching may help fortify relationships. Similarly, the fear of missing out has been associated with binge-watching. Conlin et al. [57] found that individuals may binge-watch to “catch up” on the content of dramatic series to enable participation in cultural conversations. Individuals may feel under pressure for fear of being ostracized in future conversations if they do not binge-watch a series.

In addition to the Uses and Gratification Theory, prior evidence shows that individuals with neurotic personality features may be particularly prone to binge-watching [58,59]. These findings suggest that individuals who are more likely to binge-watch may have tendencies toward experiencing and managing negative emotions, which may exacerbate mental health problems [1]. Moreover, binge-watching has been considered as a type of addictive behavior [39,60,61]. Such excessive use may worsen mental health problems; for example, individuals may sacrifice sleep or important relationships to binge-watch. After binge-watching, individuals may feel regret, distress, and unhappiness [2,62,63]. As a result, binge-watching may be associated with multiple mental health problems, especially anxiety and stress, as reflected in the present meta-analytic results.

The present systematic review and meta-analysis further revealed that associations between binge-watching and several mental health concerns including depression, loneliness, and sleep problems appeared stronger during the COVID-19 pandemic compared to before the pandemic. However, the association between binge-watching and anxiety was slightly reduced (but not statistically significantly) during the COVID-19 pandemic compared to the pre-COVID-19-pandemic period. Speculatively, the findings may in part reflect the isolation policies (e.g., quarantine and lockdown), as many governments worldwide applied isolation policies to control the spread of disease [9,10,11]. However, such policies prohibited individuals from many social interactions. Under these circumstances, individuals had limited entertainment choices, and binge-watching may have become one option for social interactions during isolation periods. However, the psychological needs to interact with friends may not have been fully satisfied when engaging in binge-watching. Speculatively, they may have experienced feelings of loss after binge-watching if they could not discuss the content with their friends in person. These circumstances may have worsened mental health and resulted in stronger associations between binge-watching and mental health concerns during the COVID-19 pandemic. This may also explain why associations became stronger for depression, loneliness, and sleep problems but not anxiety. That is, limited interactions with friends may have been less likely to generate anxiety, but more likely to produce loneliness, depression, and sleep problems. However, data on the associations between binge-watching and mental health problems during the COVID-19 pandemic remain scarce (i.e., only five papers were reviewed in the present systematic review and meta-analysis). Therefore, future studies are needed to provide additional evidence regarding the associations between binge-watching and mental health problems during the COVID-19 pandemic.

Binge-watching was found to have stronger associations with stress and sleep problems in developing countries than in developed countries. Speculatively, these relationships may be in part be explained by the health care systems between developing countries and developed countries. Specifically, developed countries typically have better mental health support systems than developing countries [64]. Thus, people in developed countries may have had more health care access and care to support them and to cope with their mental health problems. Thus, people in developed countries may have been less likely to have used binge-watching to cope with their psychological needs than those in developing countries. Future studies should directly test this possibility.

The present findings have important implications. Health care providers may wish to pay more attention to binge-watching behaviors given associations with poor mental health. More specifically, it could be advantageous to understand whether people are using binge-watching to cope with the psychological difficulties they are experiencing. Then, health care providers may provide appropriate interventions for them to improve their coping skills to better address their psychological difficulties. This may minimize their binge-watching and reduce the possibilities of developing future mental health problems, particularly if binge-watching is exacerbating mental health concerns.

### Limitations

The present study had some limitations. First, the instruments assessing mental health and binge-watching varied across studies (e.g., some used self-designed instruments and others used questionnaires that have been validated; see Table 1 for details). The use of different instruments carries a risk of measuring different constructs. Therefore, the synthesized data could have been biased due to the different instruments employed. However, we have tried to minimize such bias by checking aspects including the quality of assessment instruments. Specifically, all studies reviewed and synthesized in the present systematic review and meta-analysis used a valid and reliable instrument. Therefore, such bias in using different instruments may not be substantial. Second, all studies used cross-sectional designs, precluding the drawing of causal inferences regarding relationships between binge-watching and mental health concerns. Therefore, future studies using longitudinal designs are warranted. Third, the age of the participants in the studies ranged between 18 and 68 years, inclusively. Therefore, associations between binge-watching and mental health concerns were only considered for people within this age range. In other words, findings in the present systematic review and meta-analysis may not generalize to other age groups (e.g., children and adolescents). Fourth, some studies had poor methodology quality, and thus may have provided biased information when assessing the relationships between binge-watching and mental health concerns. Nevertheless, we checked all studies for possible publication bias and considered these in the analyses and interpretations of findings as indicated. Fifth, we only searched English-written papers, therefore, the present meta-analysis may have missed some important information from studies published in other languages. Finally, we did not assess all types of mental health problems in the present study. Specifically, behavioral addictions are important mental health problems and could be associated with binge-watching. However, the present study did not assess behavioral addictions, and future studies should examine this topic.

## 5. Conclusions

In conclusion, binge-watching is a phenomenon that health care providers and policymakers should consider. Although associations between binge-watching and mental health problems were not strong in the present systematic review and meta-analysis, such associations were significant and positive. Therefore, people who binge-watch appear more likely to experience multiple mental health problems, and this may result in personal suffering and increased health care expenditures. The present meta-analysis is novel in that this is the first study to accumulate quantitative evidence in the literature to understand how binge-watching may be associated with specific mental health problems. Moreover, considering the COVID-19 pandemic in the meta-analysis is another novel aspect of the present study. Thus, it is suggested that programs and interventions focusing on reducing binge-watching (especially more extreme patterns) should be developed and empirically validated to examine the effects on mental health. Such interventions may be particularly relevant given the COVID-19 pandemic given the stronger relationships observed between binge-watching and several mental health concerns during the pandemic. Furthermore, such interventions may be particularly helpful for people in developing nations.

## Figures and Tables

**Figure 1 ijerph-19-09707-f001:**
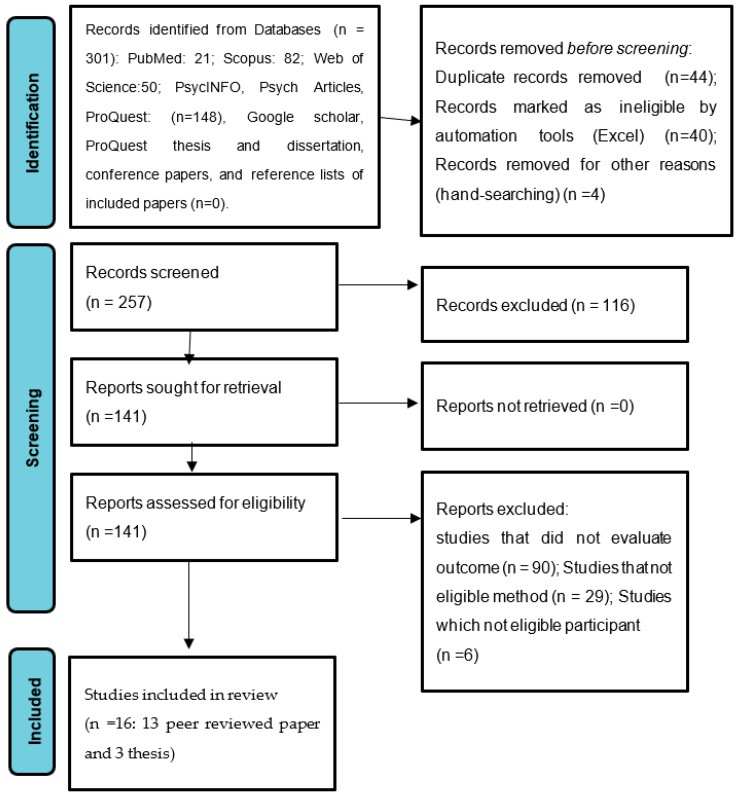
The study PRISMA flow diagram from the search to the selection of eligible studies.

**Figure 2 ijerph-19-09707-f002:**
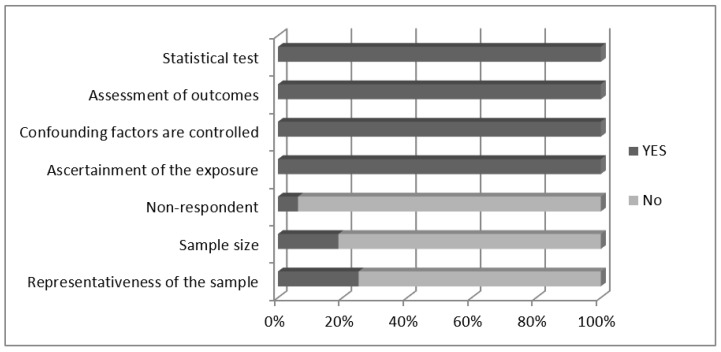
The results of the methodological quality assessment of the included studies.

**Figure 3 ijerph-19-09707-f003:**
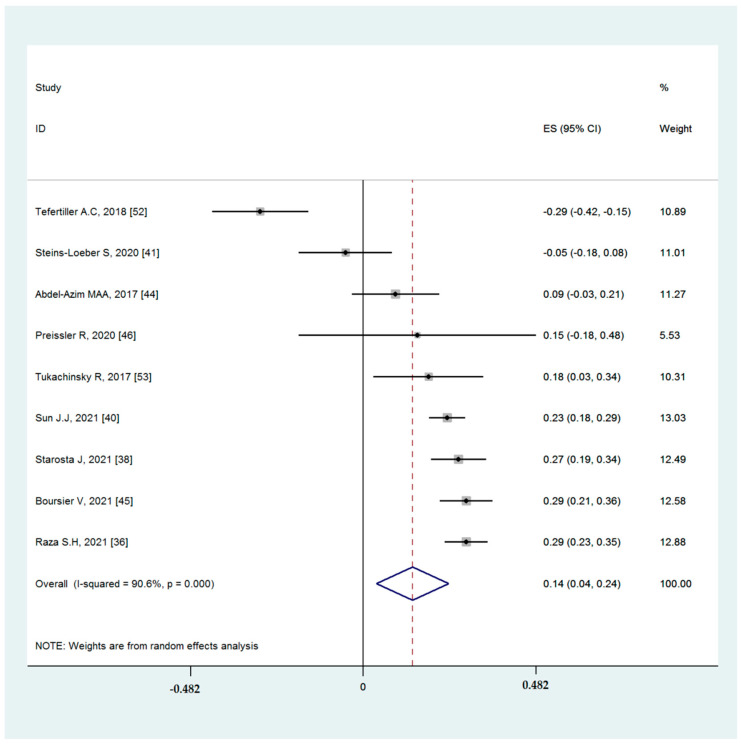
The forest plot displaying the estimated pooled Fisher’s z-score in the association between binge-watching and depression.

**Figure 4 ijerph-19-09707-f004:**
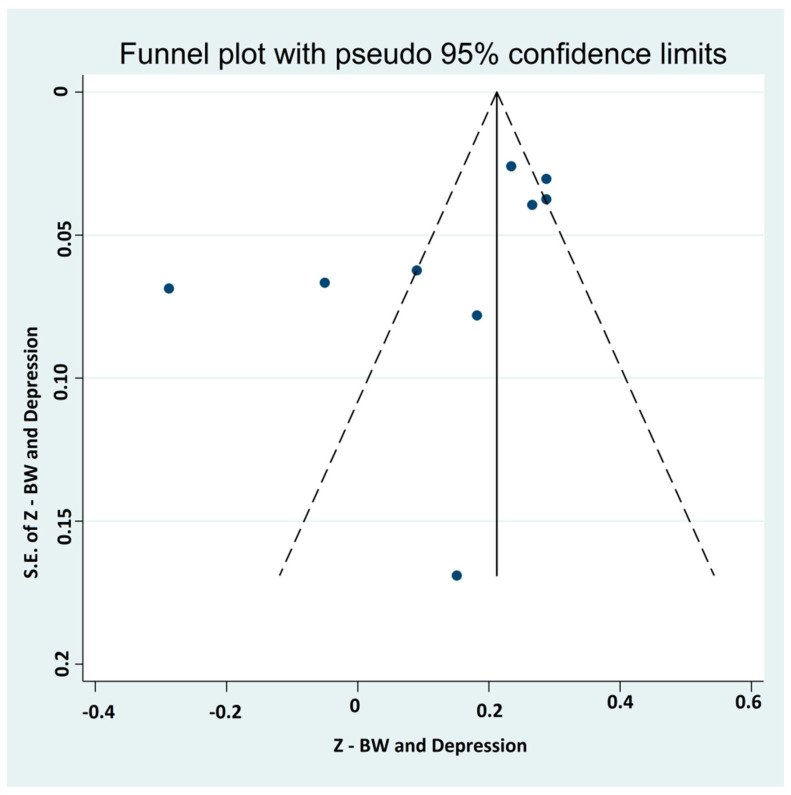
The funnel plot displaying the estimated pooled Fisher’s z-score in the association between binge-watching and depression.

**Figure 5 ijerph-19-09707-f005:**
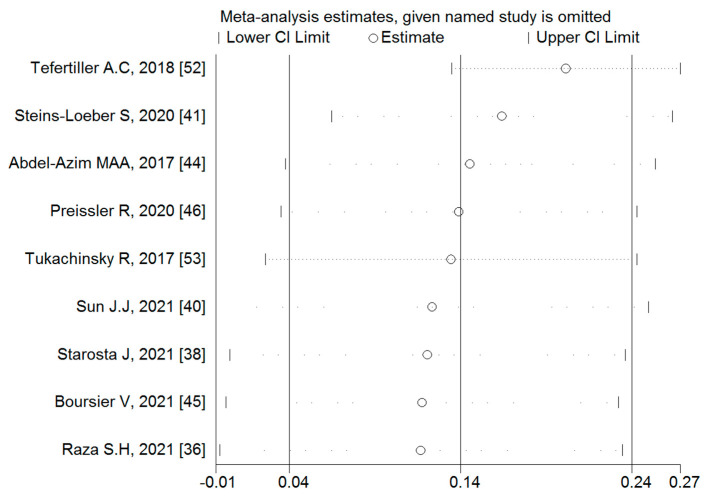
The sensitivity analysis plot assessing the small study effect in the estimated pooled Fisher’s z-score in the association between binge-watching and depression.

**Figure 6 ijerph-19-09707-f006:**
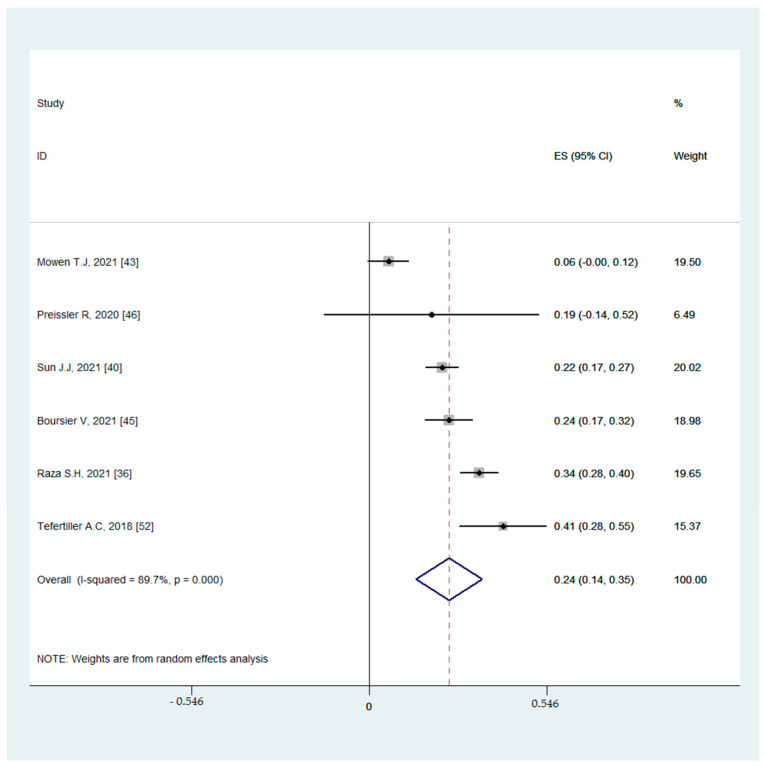
The forest plot displaying the estimated pooled Fisher’s z-score in the association between binge-watching and anxiety.

**Figure 7 ijerph-19-09707-f007:**
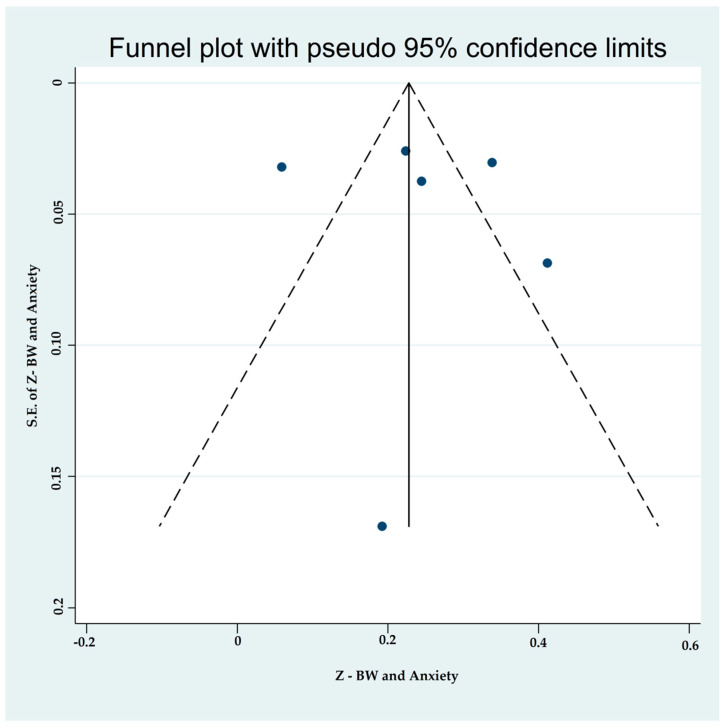
The funnel plot displaying the estimated pooled Fisher’s z-score in the association between binge-watching and anxiety.

**Figure 8 ijerph-19-09707-f008:**
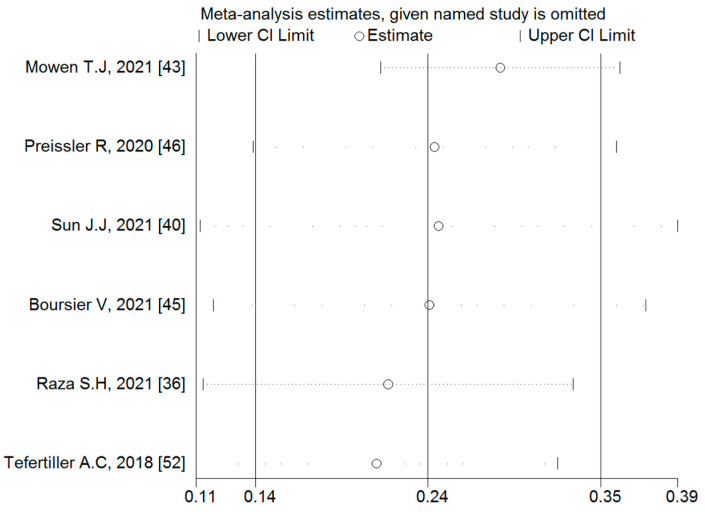
The sensitivity analysis plot assessing the small study effects in the estimated pooled Fisher’s z-score in the association between binge-watching and anxiety.

**Figure 9 ijerph-19-09707-f009:**
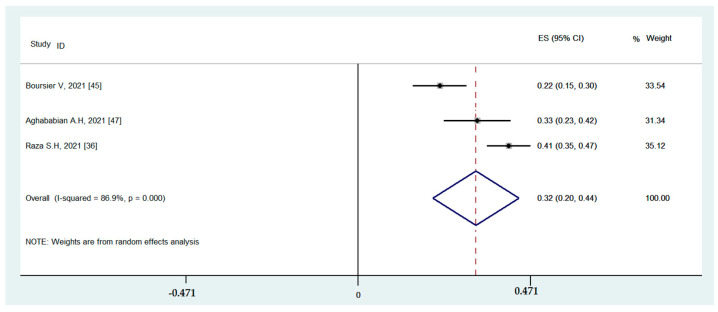
The forest plot displaying the estimated pooled Fisher’s z-score in the association between binge-watching and stress.

**Figure 10 ijerph-19-09707-f010:**
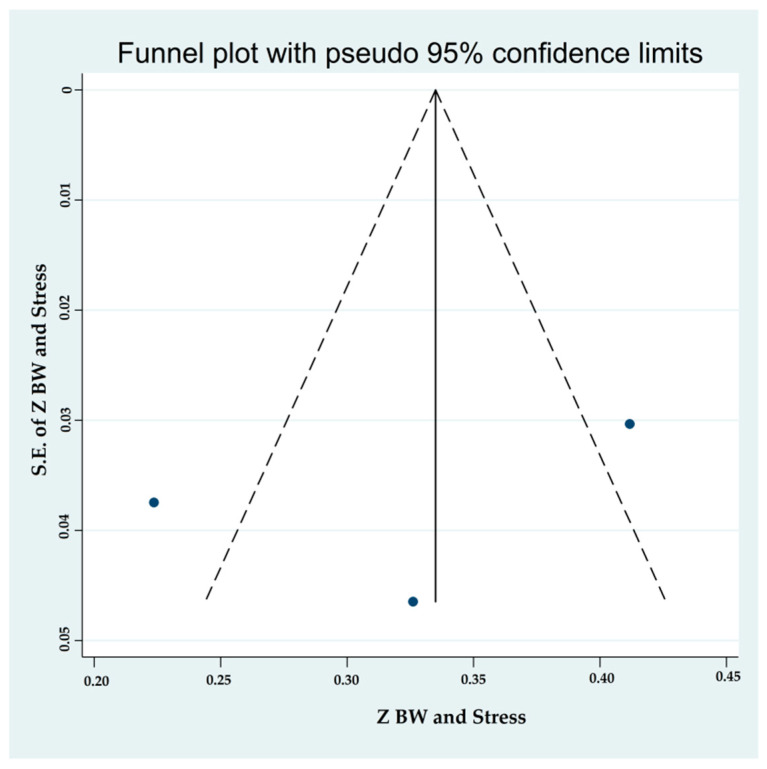
The funnel plot displaying the estimated pooled Fisher’s z-score in the association between binge-watching and stress.

**Figure 11 ijerph-19-09707-f011:**
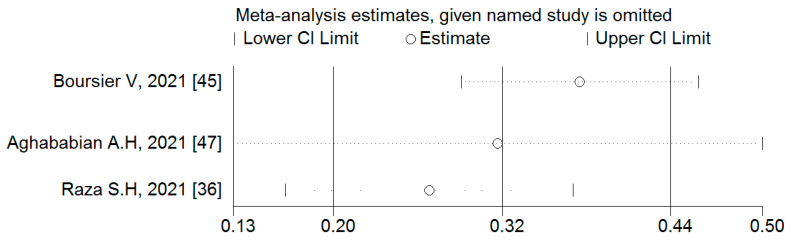
The sensitivity analysis plot assessing the small study effect in the estimated pooled Fisher’s z-score in the association between binge-watching and stress.

**Figure 12 ijerph-19-09707-f012:**
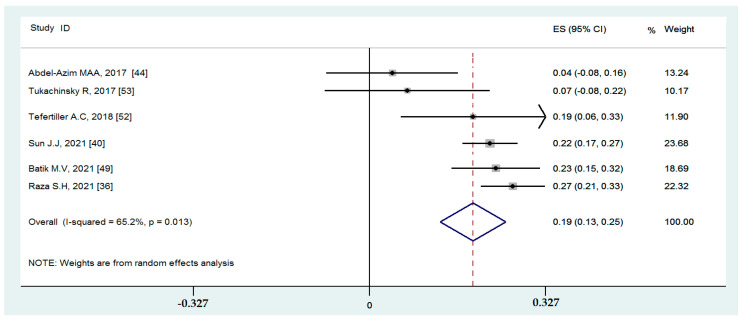
The forest plot displaying the estimated pooled Fisher’s z-score in the association between binge-watching and loneliness.

**Figure 13 ijerph-19-09707-f013:**
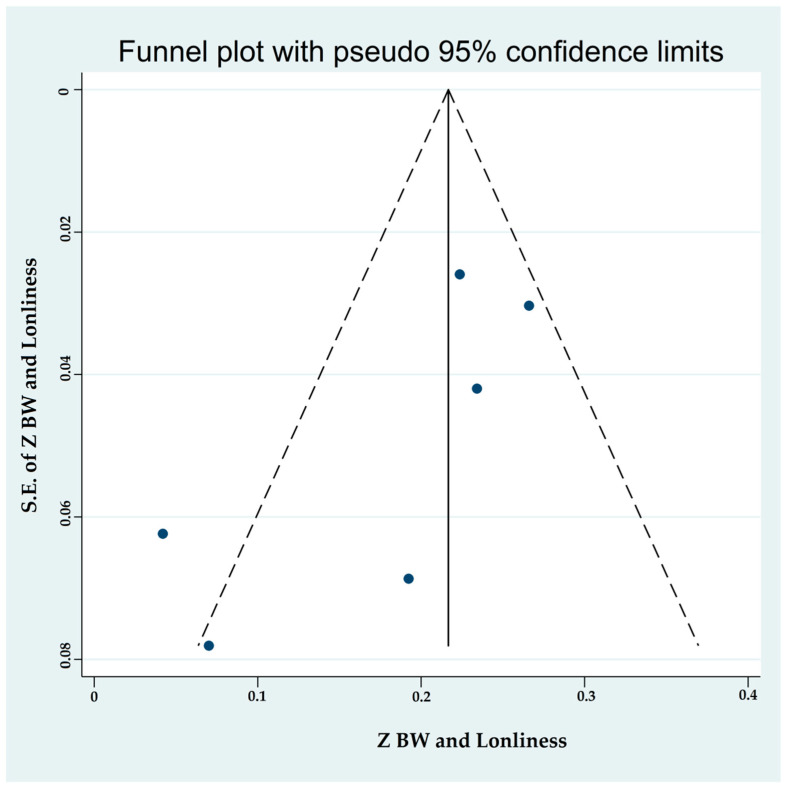
The funnel plot displaying the estimated pooled Fisher’s z-score in the association between binge-watching and loneliness.

**Figure 14 ijerph-19-09707-f014:**
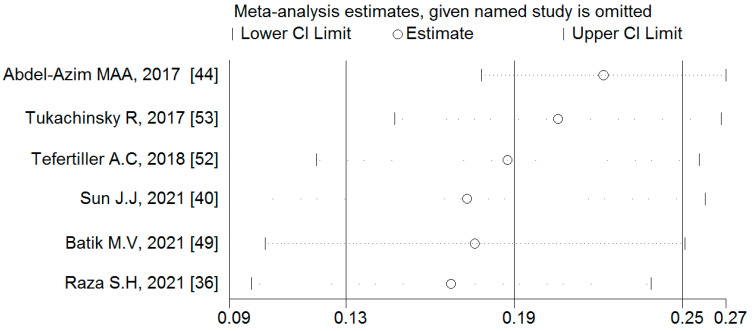
The sensitivity analysis plot assessing the small study effect in the estimated pooled Fisher’s z-score in the association between binge-watching and loneliness.

**Figure 15 ijerph-19-09707-f015:**
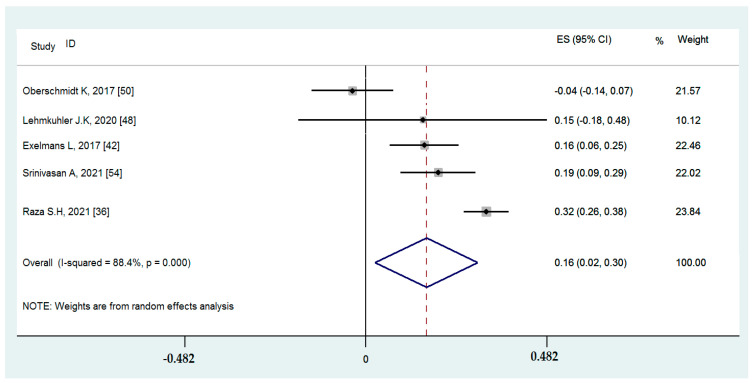
The forest plot displaying the estimated pooled Fisher’s z-score in the association between binge-watching and insomnia.

**Figure 16 ijerph-19-09707-f016:**
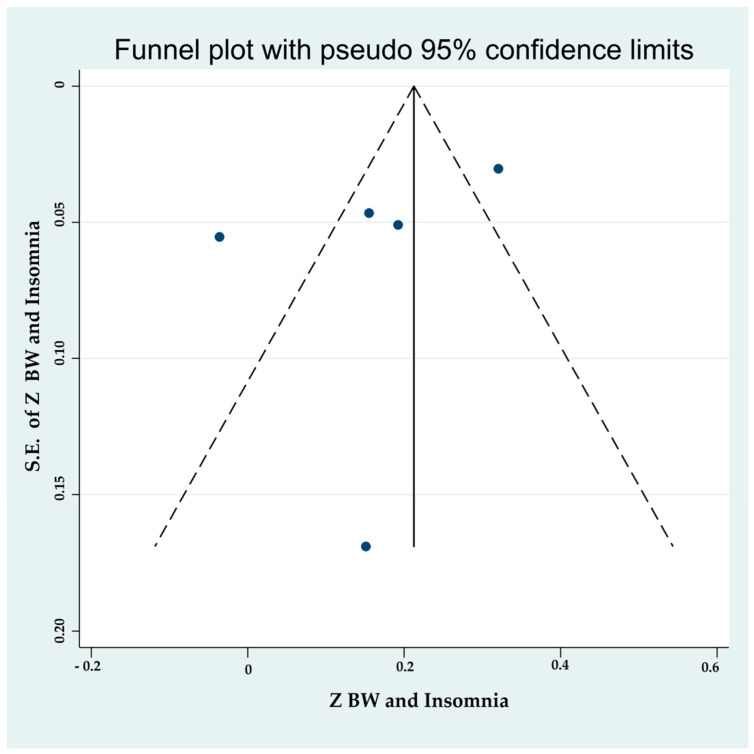
The funnel plot displaying the estimated pooled Fisher’s z-score in the association between binge-watching and insomnia.

**Figure 17 ijerph-19-09707-f017:**
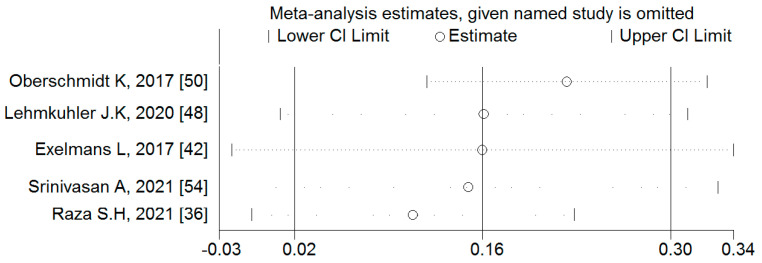
The sensitivity analysis plot assessing the small study effect in the estimated pooled Fisher’s z-score in the association between binge-watching and insomnia.

**Table 1 ijerph-19-09707-t001:** The summarized characteristics of the included studies.

First Author (Publication Year/Country)	During COVID-19 Pandemic	Participant Group	Sample Size	% Female	Age Range/ Mean Age	Binge-Watching Scale	Psychological Measurement	Score
Raza S.H. et al. (2021/Pakistan) [36]	Yes	Adult	1089	45.63	18–69	Merikivi et al. [37]	PHQ-4 (PHQ-2 and GAD-2)	6
Starosta J. et al. (2021/Poland) [38]	Yes	Adult	645	83.25	18–30	Excessive binge-watching by Starosta et al. [39]	STAI, Depression measurement questionnaire by Lojek et al.	5
Sun J.J. et al. (2021/ Taiwan) [40]	No	Adult	1488	74.86	28.3	PSWS	UCLA Loneliness Scale, CES-D, SIAS-C	5
Steins-Loeber S. et al. (2020/Germany) [41]	No	Adult	228	64.91	18–58	BWESQ	ADS-K	6
Exelmans L. et al. (2017/Belgium) [42]	No	Adult and adolescence	463	61.98	18–25	Researcher made	PSQI, BIS	5
Mowen T. et al. (2021/USA) [43]	Yes	University student	957	69.28	20.4	Researcher made	Multidimensional Assessment of Anxiety	6
Abdel-Azim Mohamed Ahmed A. et al. (2017/Arab Emirates) [44]	No	Adult	260	52.31	18–48	Researcher made	CESD, UCLA Loneliness	6
Boursier V. et al. (2021/Italy) [45]	Yes	Adult	715	71.46	18–72	BWESQ	DASS-21	5
Preissler R. et al. (2020/Netherland) [46]	No	Adult	38	44.73	23.7	Researcher made	PHQ, GAD	7
Aghababian A.H. et al. (2021/USA) [47]	Yes	Adult	466	63.73	Over 18	Researcher made	Covid-Related Stress (Researcher made)	5
Lehmkühler J.K. et al. (2020/Netherland) [48]	NO	Adult	38	44.73	23.7	Researcher made	PHQ, GAD	5
Batik M.V. et al. (2021/Turkey) [49]	NO	Adult	570	410	18–68	BWESQ	UCLA Loneliness Scale	5
Oberschmidt K. et al. (2017/Netherland) [50]	NO	Adult	329	72.03	17–33	Walton-Pattison et al. [51]	CHB scale	5
Tefertiller A.C. (2018/USA) [52]	NO	Adult	215	46.04	36	Researcher made	STAI, Depression measurement questionnaire by Norris and Mitchell (2014)	5
Tukachinsky R. (2017/USA) [53]	NO	Student	167	80.83	22–46	Researcher made	Depression measurement questionnaire by Mirowsky and Ross (1992), UCLA Loneliness Scale	5
Srinivasan A. (2021/ India) [54]	No	College Students	391	60.6	NR	Researcher made	PSQI	6

STAI: State-Trait Anxiety Inventory, CES-D: Center for Epidemiologic Studied Depression Scale, SIAS-C: Social Interaction Anxiety Scale, PSWS: Problematic Series Watching Scale, BWES: Binge-Watching Effect Scale, PSQI: Pittsburgh Sleep Quality Index, BIS: Bergen Insomnia Scale, BWESQ: Binge-Watching Engagement and Symptoms Questionnaire, DASS-21: Depression Anxiety Scale, CHB: Compensatory Health Beliefs, NR: Not reported.

**Table 2 ijerph-19-09707-t002:** The pooled effect sizes for associations between binge-watching and different mental health problems.

	Number of Studies	Fisher’s z-Score	95% CI	I^2^ (%)	Tau^2^
Depression	9	0.14	0.04 to 0.24	90.6	0.02
Anxiety	6	0.25	0.14 to 0.35	89.7	0.01
Stress	3	0.32	0.20 to 0.44	86.9	0.01
Loneliness	6	0.19	0.13 to 0.25	65.2	0.01
Insomnia	5	0.16	0.03 to 0.30	88.4	0.02

**Table 3 ijerph-19-09707-t003:** The subgroup analyses for the associations between binge-watching and mental health problems.

Mental Health Problem (n)			No. of Studies	ES (95% CI)	I^2^ (%)	Tau^2^
Depression (n = 9)	COVID-19 pandemic	Yes	3	0.28 (0.24; 0.32)	0	0
		No	6	0.05 (−0.13; 0.22)	91.8	0.04
	Risk of bias	Low	4	0.12 (−0.06; 0.31)	88.3	0.03
		High	5	0.15 (−0.001; 0.29)	93.3	0.03
	Country developmental status	Developing	2	0.20 (0.004; 0.39)	87.7	0.02
		Developed	7	0.12 (−0.02; 0.25)	92	0.03
Anxiety (n = 6)	COVID-19 pandemic	Yes	3	0.21 (0.05; 0.384)	95.1	0.02
		No	3	0.29 (0.14; 0.44)	70	0.01
	Risk of bias	Low	3	0.20 (−0.04; 0.43)	95	0.04
		High	3	0.27(0.19; 0.36)	69.6	0.003
	Country developmental status	Developing	1	0.34 (0.28; 0.40)	-	-
		Developed	5	0.22 (0.11; 0.33)	87.1	0.01
Stress (n = 3)	COVID-19 pandemic	Yes	3	0.32 (0.20; 0.44)	86.9	0.009
	Risk of bias	Low	1	0.41 (0.35; 0.47)	-	-
		High	2	0.27 (0.17; 0.37)	66	0.004
	Country developmental status	Developing	1	0.41 (0.35; 0.47)	-	-
		Developed	2	0.27(0.17;0.37)	66	0.004
Loneliness (n = 6)	COVID-19 pandemic	Yes	1	0.27 (0.21; 0.33)	-	-
		No	5	0.17 (0.10; 0.24)	62.5	0.004
	Risk of bias	Low	2	0.16 (−0.06; 0.38)	90.4	0.02
		High	4	0.21 (0.16; 0.26)	22.3	0.001
	Country developmental status	Developing	2	0.16 (−0.06; 0.38)	90.4	0.02
		Developed	4	0.21 (0.16; 0.26)	22.3	0.001
Insomnia (n = 5)	COVID-19 pandemic	Yes	1	0.32 (0.26; 0.38)	-	-
		No	4	0.11 (−0.01; 0.23)	71.5	0.01
	Risk of bias	Low	2	0.26 (0.14; 0.39)	78.6	0.01
		High	3	0.08 (−0.08; 0.23)	72.1	0.01
	Country development status	Developing	2	0.26 (0.14; 0.39)	78.6	0.007
		Developed	3	0.08 (−0.08; 0.23)	72.1	0.01

**Table 4 ijerph-19-09707-t004:** The results of the meta-regression analysis in association between binge-watching and mental health problems.

		Univariate Regression							Multivariate Regression						
Mental Health Problems	Variables	No. of Studies	Coeff.	S.E.	*p*	I^2^ Residual (%)	Adj. R^2^ (%)	Tau^2^	No. of Studies	Coeff.	S.E.	*p*	I^2^ Residual (%)	Adj. R^2^ (%)	Tau^2^
Depression (n = 9)	Country	9	−0.02	0.03	0.51	91.05	−6.27	0.04	6						
	NOS score	9	−0.01	0.11	0.93	91.79	−13.26	0.04							
	COVID−19 pandemic (Yes vs. No)	9	0.23	0.11	0.08	88.57	32.59	0.02		0.25	0.12	0.16	53.78	85.92	0.006
	Participants mean age (years)	6	−0.03	0.02	0.27	93.35	13.52	0.04		−0.04	0.01	0.11			
	Percentage of female participants	9	0.005	0.004	0.24	91.35	8.55	0.03		0.008	0.004	0.19			
Anxiety (n = 6)	Country	6	−0.01	0.02	0.59	89.24	−16.59	0.02	5						
	NOS score	6	−0.07	0.09	0.5	91.42	−8.75	0.01							
	COVID-19 pandemic (Yes vs. No)	6	−0.07	0.11	0.55	91.57	−15.34	0.02							
	Participants mean age (years)	5	0.02	0.004	0.01	0	100	0		0.19	0.004	0.04	0	99.02	0.0001
	Percentage of female participants	6	−0.006	0.004	0.19	86.75	31.02	0.01		−0.002	0.002	0.58			
Stress (n = 3)	Country	3	−0.03	0.007	0.16	0	100	0	Number of studies were insufficient						
	NOS score	3	0.14	0.08	0.34	66.03	56.68	0.003							
	Percentage of female participants	3	−0.007	0.002	0.16	0	100	0							
Loneliness (n = 6)	Country	6	−0.01	0.01	0.47	66.01	−19.81	0.01	As no significant predictor was found in the univariable meta-regression, multivariable analysis was not conducted.						
	NOS score	6	−0.02	0.09	0.83	72.02	−47.82	0.007							
	COVID-19 pandemic (Yes vs. No)	6	0.1	0.09	0.31	62.46	1.75	0.005							
	Participants mean age (years)	4	0.11	0.01	0.46	70.06	−18.80	0.005							
	Percentage of female participants	6	−0.001	0.003	0.8	71.26	−43.65	0.007							
Insomnia (n = 5)	Country	5	−0.02	0.02	0.29	80.08	19.79	0.01	Number of studies were insufficient						
	NOS score	5	0.19	0.1	0.16	74.66	46.78	0.01							
	COVID-19 pandemic (Yes vs. No)	5	0.21	0.12	0.18	71.49	42.89	0.01							
	Participants mean age (years)	3	0.1	0.05	0.3	18.86	86.91	0.001							
	Percentage of female participants	3	−0.01	0.002	0.01	-	100	0							

Coeff: Coefficient; S.E.: Standard Error; NOS score: methodological quality score based on Newcastle–Ottawa checklist; Adj. R^2^: Adjusted R^2^.

## Data Availability

All data are included in the paper.

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
