# Peer review of "Binge-Watching and Mental Health Problems: A Systematic Review and Meta-Analysis"

_ijerph, 2022, doi:10.3390/ijerph19159707_

Round 1
Reviewer 1 Report
The article consists of a meta-analysis that deals with binge-watching and mental health problems.
It has been carried out following PRISMA regulations and its protocol has been registered on the PROSPERO website.
In general I have observed some misspellings due to missing letters in a word.
The abstract fully records the methodology used, the results obtained and the conclusions correctly.
In the methodology section, although they point out that it follows the PRISMA standard, I miss certain sections that are missing and order in them.
Not having carried out the search in different languages ​​produces a bias that must be pointed out.
The quality of the articles is low, is it taken into account in limitations?
The flowchart could be improved, sometimes the values ​​of n do not seem clear.
The bibliography could be improved, several of the citations do not have the correct page numbers, in some the publisher of the book is missing and in others the data of the magazine.
Author Response
- The article consists of a meta-analysis that deals with binge-watching and mental health problems.
It has been carried out following PRISMA regulations and its protocol has been registered on the PROSPERO website.
Response: Thank you for the positive comment. We also deeply appreciate your following constructive comments, which guide us in improving our work.
- In general I have observed some misspellings due to missing letters in a word.
Response: We are sorry for the misspellings. We have now carefully checked through the manuscript to identify and remedy the typographical errors.
- The abstract fully records the methodology used, the results obtained and the conclusions correctly.
Response: Thank you.
- In the methodology section, although they point out that it follows the PRISMA standard, I miss certain sections that are missing and order in them.
Response: Thank you for the recommendation. We have now organized the Methods section to cover all PRISMA sections.
- Not having carried out the search in different languages produces a bias that must be pointed out.
Response: Thank you for this comment. We have now addressed this as one of the limitations.
“Fifth, we only searched English-written papers; therefore, the present meta-analysis might miss some important information from studies published in other languages.”
- The quality of the articles is low, is it taken into account in limitations?
Response: We agree. We have now mentioned this as one of the limitations.
“Fourth, some studies had poor methodology quality, and thus may have provided biased information when assessing relationships between binge-watching and mental health concerns.”
- The flowchart could be improved, sometimes the values of n do not seem clear.
Response: We have now improved the quality of the flowchart and have striven to clarify the numbers.
- The bibliography could be improved, several of the citations do not have the correct page numbers, in some the publisher of the book is missing and in others the data of the magazine.
Response: Thank you for identifying this concern for us. We have now rechecked all the citations to ensure their accuracy and appropriateness.
Reviewer 2 Report
Current report investigated the associations between binge-watching and five mental health concerns, including depression, loneliness, sleep problems, anxiety, and stress. Please conduct the concerns below.
1. Merit(s) of systematic review with a meta-analysis for binge-watching must introduce in clear, particularly the published reference(s).
2. Please check the Reference 21.
3. Members of the review team searched academic databases were not indicated.
4. The longitudinal, case-control and cross-sectional studies must introduce in clear.
5. Duration of the COVID-19 pandemic in current analysis was unknown.
6. The 16 included studies did not include Japan or Korea. Why? Taiwan seems used to represent as the Asian area?
7. Bias in synthesized data belonged to the limitations of current analysis. How to rule out of it? Please discuss in detail.
8. Novelty was not indicated in the conclusions.
Author Response
Current report investigated the associations between binge-watching and five mental health concerns, including depression, loneliness, sleep problems, anxiety, and stress. Please conduct the concerns below.
Response: Thank you for the general comment and the following constructive comments. We have now improved our work according to your suggestions.
- Merit(s) of systematic review with a meta-analysis for binge-watching must introduce in clear, particularly the published reference(s).
Response: We have now more clearly described the merits as follows:
“The merits of the present meta-analysis include using published references to understand associations between binge-watching and mental health problems. This is particularly important given changes in technology that have facilitated binge-watching behaviors and mental health concerns that may relate to screen media activity and specific forms like binge-watching. Moreover, meta-analyses quantify and accumulate existing evidence and can provide clear and robust information regarding associations.”
- Please check the Reference 21.
Response: We have now updated the Reference 21.
“Pakpour A, Jafari E. The associations of problematic binge-watching with mental health including depression, anxiety, and insomnia. National Institute for Health Research: PROSPERO; International prospective register of systematic reviews; 2022 [PROSPERO CRD42022309716]. Available from: https://www.crd.york.ac.uk/prospero/display_record.php?ID=CRD42022309716.”
- Members of the review team searched academic databases were not indicated.
Response: We have now added the information.
“EJ and ZA contributed in developing search syntax and conducting the search of academic databases.”
- The longitudinal, case-control and cross-sectional studies must introduce in clear.
Response: We have now provided definitions for longitudinal, case-control, and cross-sectional studies.
“Observational studies including longitudinal (i.e., having two or more assessment time points in the study), case-control (i.e., having one group with binge-watching concerns and another group without in the study) and cross-sectional studies (i.e., having only one assessment time point in the study) published in English through February of 2022 were included if relevant data regarding binge-watching and associations with selected mental health problems (depression, loneliness, sleep problems, anxiety, and stress) were reported.”
- Duration of the COVID-19 pandemic in current analysis was unknown.
Response: Thank you for the reminder. We have now clearly defined the time that is considered under the COVID-19 pandemic.
“Specifically, we defined a study as having been conducted during the COVID-19 pandemic when its data collection time was after December of 2019.”
- The 16 included studies did not include Japan or Korea. Why? Taiwan seems used to represent as the Asian area?
Response: Studies were included that were published in English and had the necessary data. Considering that the search was systematic, it cannot be said why a study from a country was included or not. It is possible that the lack of inclusion of studies from these countries is a limitation of the study given that there were no studies identified using the defined search criteria. This situation may reflect language (that is, the search identified studies published in English and it is possible that there existed studies published in other languages such as Japanese or Korean). We have noted the language limitation of the present study. Nonetheless, Taiwan is located in East Asia, and thus we considered it as part of Asia in the current study.
“Fifth, we only searched English-written papers; therefore, the present meta-analysis may have missed some important information from studies published in other languages.”
- Bias in synthesized data belonged to the limitations of current analysis. How to rule out of it? Please discuss in detail.
Response: Thank you. We have now described this point in greater detail.
“Therefore, the synthesized data could have been biased due to the different instruments employed. However, we have tried to minimize such bias via checking aspects including the quality of assessment instruments. Specifically, all studies reviewed and synthesized in the present systematic review and meta-analysis used a valid and reliable instrument. Therefore, such bias in using different instruments may not be substantial.”
“Nevertheless, we have checked all studies for publication bias and considered this in analysis and interpretation.”
- Novelty was not indicated in the conclusions.
Response: Thank you. We have now mentioned the novelty of the present study in the conclusions.
“The present meta-analysis is novel in that this is the first study to accumulate quantitative evidence in the literature to understand how binge-watching may be associated with specific mental health problems. Moreover, considering the COVID-19 pandemic in the meta-analysis is another novel aspect of the present study.”

Reviewer 3 Report
Title
• In my opinion, In the title, it should be stated that this work is a meta-analysis only. In the first place, if it is a meta-analysis, the process of searching and selecting articles should have been systematic, so it is not necessary to indicate this again in the title. It is redundant. Secondly, I would understand that the authors included “Systematic Review” in the title if, in addition to a quantitative synthesis (meta-analysis), the authors had done a qualitative synthesis, but this is not the case.
Introduction
• I lack an explanation of the relationship between the variables from a theoretical approach. I see this is under discussion. Perhaps it could be named in the introduction and developed in the discussion.
Study aims
• The authors select mental health problems to address: depression, loneliness, sleep problems, anxiety, and stress. In the introduction, as a result of a previous review, “potential health risks resulting from binge-watching include developing behavioral addictions, sleep problems, sedentary behaviors, and psychological distress (e.g., depression, anxiety, loneliness, and stress)” are indicated. This research includes depression, loneliness, sleep problems, anxiety, and stress, but, Why areaddictions not included as a mental health problem?
Method
Study screening and selection process
• Finally, the authors did not include any references from informal methods. I think the authors should report this in “Study screening and selection process” and add it to the flowchart. This is to make it transparent that the authors did indeed include informal methods, albeit without success.
Study description
• I think the authors should include information on the countries in the synthesis of the description of the studies. I think it's interesting that readers have a clearer idea of ​​where this issue is addressed. For example, no study was conducted in a Latin American country.
Data synthesis
• The authors do not explain the Subgroup analysis of the meta-regression when the rest of the analyzes explain them in detail. What is I2 res? What is Tau2?
Results
Associations between binge-watching and depression
• The authors include in Table 3 a variable that has not been discussed previously "Country development status." How was this variable calculated? Why is it included?
• The authors only comment on the results of the work carried out during the pandemic or before, but they do not comment on the results of the other two variables listed in Table 3. Either these analyzes are eliminated, or they are commented on.
• The authors start talking about univariate meta-regression analysis at line 226, but it is not until the end of the paragraph that they indicate the table they are talking about. I think they should announce it sooner.
• Table 4 is confusing and does not meet APA 7 criteria. Perhaps, it could help to put the page horizontally. It might also help to include the variables' acronyms, explaining them in a note at the bottom of the table.
• Perhaps there is something I am not understanding. The authors say, "Based on univariate meta-regression analysis, during the pandemic, participants' mean age (in years) and female gender were statistical predictors with respect to the association between binge-watching and depression." But the p values ​​for the variables "mean age (in years)" and "female gender" are greater than .05, so this result should not be significant. The same is true for other outcomes for other mental health problems, such as anxiety, stress, and insomnia.
Associations between binge-watching and stress
• No reference is made to results in Table 3.
Discussion
I liked that the authors talked about using different instruments to measure the different variables as a limitation. I agree that this is the most critical limitation of the article. Likewise, I think it would be interesting if the authors recognized that using different instruments implies measuring various constructs.
Author Response
Title
- In my opinion, In the title, it should be stated that this work is a meta-analysis only. In the first place, if it is a meta-analysis, the process of searching and selecting articles should have been systematic, so it is not necessary to indicate this again in the title. It is redundant. Secondly, I would understand that the authors included “Systematic Review” in the title if, in addition to a quantitative synthesis (meta-analysis), the authors had done a qualitative synthesis, but this is not the case.
Response: Thank you for the explanation. However, we consider that systematic review is a needed for conducting meta-analysis. Moreover, the PRISMA guideline indicates that it is necessary to include a systematic review and a meta-analysis at the same time. Therefore, we prefer retaining the “systematic review” in title.
Introduction
- I lack an explanation of the relationship between the variables from a theoretical approach. I see this is under discussion. Perhaps it could be named in the introduction and developed in the discussion.
Response: Thank you. We have now explained why the variables may be associated using a theoretical framework in both the Introduction and Discussion.
“Such associations appear supported by the Uses and Gratification Theory. Specifically, people may binge-watch to satisfy their mental health needs [1]. Additionally, the Uses and Gratification Theory indicates that people may receive gratification, particularly if the activity is planned [3]. However, binge-watching may be an unplanned behavior (i.e., individuals attempt not to watch too much but have difficulty ceasing the behavior). Therefore, such unplanned binge-watching may result in regret and worsening of mental health concerns [3]. As such, binge-watching may constitute a specific form of problematic use of the internet, consistent with the interaction of person, affect, cognitive and execution (I-PACE) theory.”
“These relationships resonate with the Uses and Gratification Theory [31, 32]. That is, individuals may use media to satisfy their individual needs, including psychological ones. Therefore, individuals may binge-watch to cope with troublesome and disturbing life experiences that may generate stressful and anxious feelings. Individuals may also encourage their friends to watch the same series, and such binge-watching may help them fortify relationships. Similarly, fear of missing out has been associated with binge-watching. Conlin et al. [33] found that individuals may binge-watch to “catch up” on the content of dramatic series to enable participation in cultural conversations. Individuals may feel under pressure for fear of being ostracized in future conversations if they do not binge-watch series.”
Study aims
- The authors select mental health problems to address: depression, loneliness, sleep problems, anxiety, and stress. In the introduction, as a result of a previous review, “potential health risks resulting from binge-watching include developing behavioral addictions, sleep problems, sedentary behaviors, and psychological distress (e.g., depression, anxiety, loneliness, and stress)” are indicated. This research includes depression, loneliness, sleep problems, anxiety, and stress, but, Why are addictions not included as a mental health problem?
Response: Thank you and we agree that assessing addictions could provide important information for healthcare providers or researchers. However, because addictions are too broad with different forms (e.g., internet addiction, social media addiction, gaming addiction, gambling), we consider that including addictions in the present meta-analysis would potentially burden readers. Specifically, we believe that such coverage would result in too much information being reported in one paper. As a result, we consider that this issue (i.e., how binge-watching may relate to addictions)is beyond the scope of the present review and meta-analysis and should be examined separately in another article. Nevertheless, we acknowledge the importance of this issue and have listed this as one of the limitations to be addressed in future studies.
“Lastly, we did not assess all types of mental health problems in the present study. Specifically, behavioral addictions are important mental health problems and could be associated with binge-watching. However, the present study did not assess behavioral addictions, and future studies should examine this topic.”
Method
Study screening and selection process
- Finally, the authors did not include any references from informal methods. I think the authors should report this in “Study screening and selection process” and add it to the flowchart. This is to make it transparent that the authors did indeed include informal methods, albeit without success.
Response: Thank you for the reminder. We have now added the following information.
“No references were added after conducting search in Google scholar, ProQuest thesis and dissertation, conference papers, and reference lists of included papers.”
Study description
- I think the authors should include information on the countries in the synthesis of the description of the studies. I think it's interesting that readers have a clearer idea of where this issue is addressed. For example, no study was conducted in a Latin American country.
“The 16 included studies (4 from the USA, 3 from the Netherlands, and 1 from each of the following regions: Pakistan, Poland, Taiwan, Germany, Belgium, United Arab Emirates, Italy, Turkey, and India) involved 8,077 total participants.”
Data synthesis
- The authors do not explain the Subgroup analysis of the meta-regression when the rest of the analyzes explain them in detail. What is I2 res? What is Tau2?
Response: Thank you. We have now added the information.
“I2 Index estimated heterogeneity across included studies. Heterogeneity was interpreted as mild with I2 < 25%, moderate with 25 < I2< 50%, severe with 50< I2< 75%, and highly severe with I2 > 75%”
“For assessing moderator variables and finding the main source of heterogeneity, subgroup analysis or meta-regression was conducted. Subgroup analysis is arguably the most widely used method to examine sources of heterogeneity in meta-analyses and involves the division of studies into two or more subgroups based on selected variables. Significant differences in pooled effect sizes among these subgroups are then examined [31, 32]. Meta-regression is arguably a more sophisticated method than subgroup analysis for exploring heterogeneity and has the potential advantage of efficiently allowing the evaluation of one or more covariates simultaneously [33]. Two indicators are assessed in meta-regression including the tau-square value and I2 residual. Tau-squared (τ2 or Tau2) is an estimate of the between-study variance in a random-effects meta-analysis [34]. I2 residual is an indicator showing the effect of a selected variable on observed heterogeneity. Higher values of residual variance indicate that the examined variable is not an important source of heterogeneity [35]. Notably, the threshold for deciding on the significance of a p-value in meta-regression depends on the number of studies: when there are fewer than 10 studies, the threshold is 0.20; between 10 and 20 studies, the threshold is 0.15; and above 29 studies, the threshold is 0.10 [25, 33].”
Results
Associations between binge-watching and depression
- The authors include in Table 3 a variable that has not been discussed previously "Country development status." How was this variable calculated? Why is it included?
Response: We have now provided the information of country development status and an explanation.
“Data items included first author, year of publication, study design, country, country developmental status (based on latest world bank data), sample size, target population, number of men and women, names of measurements for mental health and binge-watching, age range or mean age and effect size estimate of binge-watching and mental health. We included the variable of country developmental status in the meta-analysis because countries at different levels of development may exhibit different patterns of binge-watching behaviors and mental health problems (e.g., mental health problems may be more serious in developing countries than in developed countries because developing countries may not have as well-established mental health care systems).”
- The authors only comment on the results of the work carried out during the pandemic or before, but they do not comment on the results of the other two variables listed in Table 3. Either these analyzes are eliminated, or they are commented on.
Response: We have now described the two variables.
“Subgroup analysis (Table 3) showed that association between binge-watching and depression was significantly different before and during the COVID-19 pandemic (0.05 vs. 0.28); not significantly different between low and high risk of bias (0.12 vs. 0.15); and significantly different between developing vs. developed countries (0.20 vs. 0.12).”
“Subgroup analyses (Table 3) showed that associations between binge-watching and anxiety were not significantly different before and during the COVID-19 pandemic (0.21 vs. 0.29); not significantly different between low and high risk of bias (0.20 vs. 0.27); and significantly different between developing vs. developed countries (0.34 vs. 0.22).”
“Due to a low number of total studies, subgroup analysis was not conducted for associations between binge-watching and stress.”
“Based on subgroup analysis (Table 3) and meta-regression (Table 4), none of the examined variables influenced the pooled ES or heterogeneity in the association between binge-watching and loneliness.”
“Subgroup analysis (Table 3) showed that the association between binge-watching and insomnia was significantly stronger during versus before the COVID-19 pandemic (0.32 vs. 0.11). Also, this association was higher in low vs. high risk of bias studies that were conducted in developing vs. developed countries respectively (0.26 vs. 0.08), but this difference was not statistically significant.”
- The authors start talking about univariate meta-regression analysis at line 226, but it is not until the end of the paragraph that they indicate the table they are talking about. I think they should announce it sooner.
Response: Thank you for the suggestion. We have now modified this, and the table numbers including the investigated variables have now been inserted where the regression analysis results are mentioned for the first time.
- Table 4 is confusing and does not meet APA 7 criteria. Perhaps, it could help to put the page horizontally. It might also help to include the variables' acronyms, explaining them in a note at the bottom of the table.
Response: Thank you for the great suggestion. We have now revised Table 4 in its presentation.
- Perhaps there is something I am not understanding. The authors say, "Based on univariate meta-regression analysis, during the pandemic, participants' mean age (in years) and female gender were statistical predictors with respect to the association between binge-watching and depression." But the p values for the variables "mean age (in years)" and "female gender" are greater than .05, so this result should not be significant. The same is true for other outcomes for other mental health problems, such as anxiety, stress, and insomnia. Associations between binge-watching and stress
Response: We have now explained this.
“Notably, the threshold for deciding on the significance of a p-value in meta-regression depends on the number of studies: when there are fewer than 10 studies, the threshold is 0.20; between 10 and 20 studies, the threshold is 0.15; and above 29 studies, the threshold is 0.10 [25, 33].”
Associations between binge-watching and stress
- No reference is made to results in Table 3.
Response: Due to low number of total studies, subgroup analysis was not conducted (see above response and description of edits to the text).
Discussion
I liked that the authors talked about using different instruments to measure the different variables as a limitation. I agree that this is the most critical limitation of the article. Likewise, I think it would be interesting if the authors recognized that using different instruments implies measuring various constructs.
Response: Thank you. We have now elaborated the issue that using different instruments implies measuring various constructs.
“The use of different instruments carries a risk of measuring different constructs.”

Round 2
Reviewer 2 Report
It has been revised in a good way.